# Pathophysiology of Childhood-Onset Myasthenia: Abnormalities of Neuromuscular Junction and Autoimmunity and Its Background

**Masatoshi Hayashi**

Department of Pediatrics, Uwajima City Hospital, Uwajima, Ehime 798-8510, Japan; mas_hayashi@outlook.jp

**Abstract:** The pathophysiology of myasthenia gravis (MG) has been largely elucidated over the past half century, and treatment methods have advanced. However, the number of cases of childhood-onset MG is smaller than that of adult MG, and the treatment of childhood-onset MG has continued to be based on research in the adult field. Research on pathophysiology and treatment methods that account for the unique growth and development of children is now desired. According to an epidemiological survey conducted by the Ministry of Health, Labour and Welfare of Japan, the number of patients with MG by age of onset in Japan is high in early childhood. In recent years, MG has been reported from many countries around the world, but the pattern of the number of patients by age of onset differs between East Asia and Western Europe, confirming that the Japanese pattern is common in East Asia. Furthermore, there are racial differences in autoimmune MG and congenital myasthenic syndromes according to immunogenetic background, and their pathophysiology and relationships are gradually becoming clear. In addition, treatment options are also recognized in different regions of the world. In this review article, I will present recent findings focusing on the differences in pathophysiology.

**Keywords:** autoimmunity; childhood-onset; genetic background; myasthenia gravis; neuromuscular junction; pathophysiology





## 1. Introduction

Myasthenia gravis (MG) is understood to be a neuromuscular disorder caused by an immune disturbance at the neuromuscular junction, which results in symptoms such as muscle weakness and fatigue. In 1964, Elmqvist et al. reported abnormalities in miniature endplate potential (MEPP), which they identified as a pathophysiology of the neuromuscular junction [1]. In 1973, Jim Patrick et al. demonstrated that the immunization of rabbits with the acetylcholine receptor (AChR) can induce a myasthenic state similar to that in humans [2]; as expected from clinical findings by Simpson in 1960 [3], it became clear that MG is an autoimmune disease of the neuromuscular junction.

This autoimmune disease, MG, can be divided into an ocular MG and a generalized MG based on its clinical manifestations, and it has been noted that the clinical features of MG differ between children and adults. Furthermore, it is now clear that the dysfunction of the AChR assembly that develops at the neuromuscular junction is not only caused by acquired immune abnormalities but also by congenital genetic abnormalities. What has become clear from the reports presented on these various perspectives from various regions of the world is that each ethnic group has its own slightly different pathophysiology and that treatment is carried out within the medical culture of that region. This presentation will provide an overview of this diversity from the perspective of a Japanese pediatrician who has been practicing clinically.

## 2. Epidemiological Study

Epidemiological studies on this disease have been reported for a long time, but mostly on adult MG in Europe and the United States. Grob et al. studied 1976 MG patients from 1940 to 2000 and reported that the generalized type accounted for 1730 patients (87.6%), that there were gender differences in clinical characteristics, and that most of the severe cases died within 1–2 years after onset in the early 20th century [4]. After the development of various treatment methods, the number of patients who died despite ventilatory management was reported to be 6% by Grob et al. [4], 3.5% by C Zhang et al. in China [5], and 0.26% by Yoshikawa et al. in Japan [6]. On the other hand, Oosterhuis reported that the remission rate of MG was 30% after 15 years for the ocular MG and 18% after 20 years for the generalized MG [7]. Carr et al. summarized 55 epidemiological studies reported from various countries from 1950 to 2007 [8] and found that the frequency of MG varied greatly from country to country. Dresser et al. found the incidence to be 4.1–30/million in Europe, 3–9.1/million in North America and Japan, 0.155–0.366/million in China [9], 38.8/million in Argentina [10], and 18.1/million in Korea [11]. At the same time, Carr et al. reported that the prevalence of MG had been increasing in more recent studies than in older studies [8]. Similarly, in Korea, the incidence of MG has shown a tendency to change over the years [11], and in an Italian report, both early-onset MG (EOMG) and late-onset MG (LOMG) had incidences of 14/million around 1990, but in 2007, the incidences of EOMG and LOMG were 3/million and 37/million, respectively [12]. Osserman and Genkins reported that MG in children accounts for 11% of all cases [13], and Chiu et al. reported a difference in incidence between Taiwanese and Caucasian patients [14]. In Japan, epidemiological surveys were conducted in 1973, 1987, and 2006 by the Research Group on Neurological Intractable Diseases sponsored by the Ministry of Health, Labor, and Welfare, and the prevalence per 100,000 population increased from 5.1 in 1987 to 11.8 in 2006 and 23.1 in 2018 [6,15]. This phenomenon is probably due not only to the aging of the population but also to the fact that although the disease is inherently difficult to diagnose, advances in research have made diagnosis relatively easy and treatment methods more advanced.

### 2.1. Characteristics of Childhood Onset; Frequency and Peak Age of Onset in Childhood

In 1987, Chiu et al. investigated the frequency of MG patients according to age of onset in Taiwanese and Caucasian patients and found that pediatric MG with onset at an age of 20 years or younger accounted for 42.6% in Taiwanese and 23.3% in Caucasian subjects [14]. Most pediatric MG was ocular myopathy in Taiwanese subjects, whereas there was almost no ocular MG in Caucasians; when limited to a prepubertal age of 10 years or younger, the incidence was 23.6% in Taiwanese and 3.5% in Caucasian patients, a remarkable difference. In Japan, the Research Group for the Investigation of Neurological Intractable Diseases of the Ministry of Health, Labor and Welfare conducted an epidemiological survey, and as shown in Figure 1, a reproducible peak of onset in childhood at the age of 5 years or younger was identified in addition to an adult peak in the 20- to 50-year-old age group [6,15] (Figure 1). Epidemiological studies were subsequently conducted in various parts of the world, allowing comparisons to be made on a regional basis. Surveys in Europe and the United States have not found this childhood peak [8,9,14] has been observed not only in Japan but also in China [16,17], Taiwan [18], and Korea [19], making it a characteristic of the East Asian race. Regarding where to delimit the age for discussing childhood onset, MF Finnis et al. in the UK defined childhood as 19 years of age or younger [20]; Popperud in Norway defined childhood as 18 years of age or younger [21]; Cheng-Che Chou et al. in Taiwan defined childhood as 20 years of age or younger [18]; and Gui et al. in China defined it as 14 years of age or younger [17]. It should be noted that the criteria for childhood differ among reports. In this paper, I will define childhood as 18 years of age or younger, including post-pubertal age.

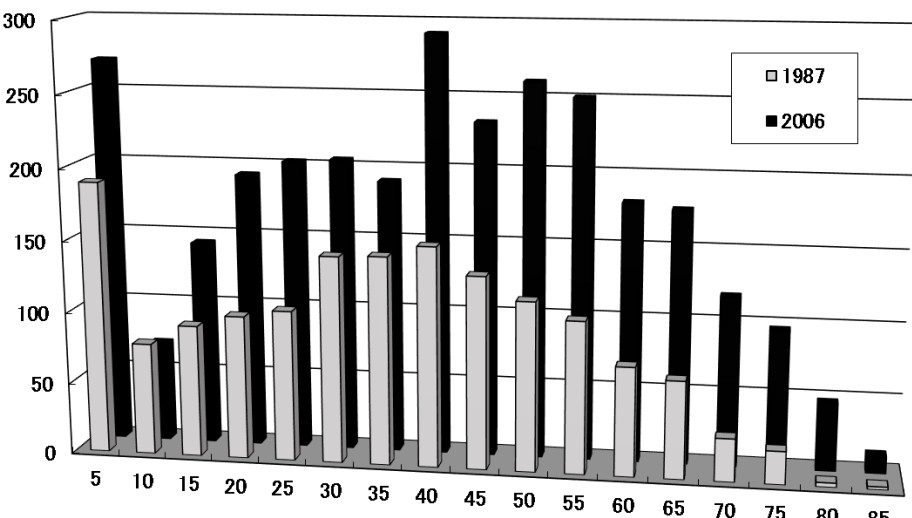

**Figure 1.** Number of MG patients by age of onset in Japan (M. Hayashi, No To Hattatsu 2022 [22]).

Another characteristic of childhood-onset MG is the ratio of ocular muscle type to generalized MG. As shown in Table 1, ocular MG accounts for a higher proportion of childhood MG in East Asia than in the West [14–19,21,23–25] (Table 1). Vecchio et al., in a study in the UK, where other ethnic groups congregate, found that the ocular muscle type in childhood-onset MG with onset before age 16 was 92% for Afro-Caribbeans, 29% for Arabians, 62.5% for Asians, and 42.3% for Caucasians, with significant racial differences among races other than Caucasians and East Asians [24]. In addition, cases initially thought to be ocular MG do not often shift to generalized MG in East Asians, even after a clinical course is followed [16–20]. In contrast, Western ocular MG has a high rate of transition to the generalized MG [9,21], and the transition often occurs within the first 2 years after onset. Thus, a comparison of pediatric MG in East Asia and the West revealed several differences.

**Table 1.** Comparison of childhood-onset MG: Ocular MG is often in Asia.

|  | Author (Reference) | Year | Nation | Number of Patients | OMG | Onset Age (yr) | OMG to GMG | Spontaneous Remission |
|---|---|---|---|---|---|---|---|---|
| Asia | Murai [15] | 2011 | Japan | 268 | 80% | <10 |  |  |
|  | Lee HN [19] | 2016 | Korea | 88 | 97% | <18 |  |  |
|  | Huang X [16] | 2013 | China | 327 | 75% | <18 | 19.9% | 3.4% |
|  | Gui [17] | 2015 | China | 424 | 83% | <14 | 11.8% |  |
|  | Yang L [25] | 2022 | China | 343 | 96% | <14 | 13.4% |  |
|  | Chou CC [18] | 2019 | Taiwan | 54 | 83% | <20 | 4.8% | 24.1% |
| Western | Popperud [21] | 2017 | Norway | 63 | 59% | <18 |  |  |
|  | Mansukhani [23] | 2019 | USA | 146 | 23% | <19 |  | 31.3% |
|  | Vecchio [24] | 2020 | UK | 74 | 51% | <16 |  | 23% |

*2.2. Immunogenetic Studies*

This racial difference in epidemiologic frequency may be due to immunogenetic differences, and several HLA differences have been reported for autoimmune MG; Vandiedonck et al. reported a strong correlation between MG with thymic hyperplasia and the 8.1 HLA haplotype [26]. HLA-A1-B8-DR3 is assigned to the 8.1 ancestral haplotype; Popperud et al. examined this in Norwegians [27] and found a strong correlation between this ancestral haplotype 8.1 (AH8.1; A*01-B*08-C*07-DRB1*03:01-DQB1*02:01), as well as alleles correlated strongly with juvenile MG. At the same time, he reported that the frequency of HLA-

B*8-DRB1*04:04 was high in early-onset MG in Europe at the age of 40 years or younger, and HLA-DRB1*15:01 alleles were high in late-onset MG at the age of 60 years or older. This was especially the case for HLA-DRB1*04:04, which was predominantly high only in prepubertal-onset cases, and it was reported that HLA types differ between older-onset and younger-onset cases [27]. High frequencies of HLA-DRB1*03 have been reported from Sweden [28], Portugal [29], and Tunisia [30], and Ancestral haplotype 8.1 is a major factor in MG development in Europe. In East Asia, HLA-A*0207, HLA-B*4601, HLA-DRB1*0403, HLA-DRB1*0901, and HLA-DRB1*1602 have been reported from China as HLAs with a high frequency in infants [31]. HLA-DR9 and DRw13 are more frequent in childhood-onset MG, and the frequency is even higher in HLA-DR9/DRw13 heterozygotes [32]. Shinomiya et al. reported that pediatric MG correlates well with HLA-DRB1*1302/DQA1*0102/DQB1*0604 and HLA-DRB1*0901/DQA1*0301/DQB1*0303 [33]. The DR13 haplotype is thought to have a close evolutionary relationship with the DR3 haplotype, which is thought to be related to Caucasian MG [34]. Although East Asia presents different HLA types from Western Europe, there may be a related immunogenetic background. On the other hand, we examined HLA in 71 Japanese MG patients, including both adults and children, and found that HLA-DRw9 was more frequent in the ocular MG and HLA-DRw8 in the generalized MG [35]. In 53 MG patients, including both children and adults, the relationship between HLA and the clinical course was investigated, and the group with HLA-DRw8 had higher AChR antibody titers with some autoantibodies. In contrast, the HLA-DRw9 group had relatively low AChR antibody titers and no autoantibodies [36]. It seems certain that a high proportion of children with MG in East Asia present with ocular MG and a high frequency of HLA-DR9 (HLA-DRB1*09;01). The relationship between these phenomena, which differ greatly between East Asia and Western Europe, and the onset of MG is an issue to be investigated in the future.

*2.3. Gender Difference*

Several studies have looked at gender differences related to the onset of MG. Murai et al. reported that the female-to-male ratio was 1.6 for cases of onset between 0 and 4 years of age, 1.5 for cases of onset between 5 and 9 years, 2.3 for cases of onset between 10 and 49 years of age, and 1.3 for cases of onset between 50 and 64 years of age [15]. According to a report by Huang et al. from China, this ratio is almost 1 for children with onset of disease under 14 years of age. Still, there are more women with onset of disease between 15 and 59 years of age, and on the contrary, more men with onset of disease after 60 years of age [16]. Furthermore, Finnis et al. reported that in childhood-onset MG, there was no difference between men and women in prepubertal cases, but this ratio was 4.5 in pubertal and 4.5 in postpubertal cases [20]. Although there are subtle differences depending on the report, there is almost no difference between men and women in the prepubertal period, but after the pubertal period, there are more women, and as they grow older, there are more men.

In 1966, Vincent P. Perlo et al. studied the frequency according to age of onset in 1355 MG patients and reported that there was a young female peak and an elder male peak, two peaks that differed according to sex [37]. In a systematic review, Carr et al. also showed a clear difference in age of onset via a beautiful figure [8]. Similarly, surveys in Japan have shown a pattern of different peak incidences in men and women [6,15], but with each passing year, life expectancy increases and the pattern of gender differences becomes harder to discern. In 1980, Compston et al. added HLA analysis and reported that the younger female group had higher AChR antibody titers; more HLA-A1, B8, and/or DRw3; and more thymic hyperplasia, whereas the elder male group had lower antibody titers and more HLA-A3, B7, and/or DRw2 [38]. Ancestral haplotype 8.1, which is more common in younger females, may be associated with the fact that various autoimmune diseases are more frequent in women of this age.

### 3. Pathophysiology of Myasthenic State

The pathophysiology of childhood MG, especially MG with post-pubertal onset, is basically similar to that of adults. In some cases, such as neonatal transient MG and congenital joint contractures, there is a pathophysiology specific to children, and there are problems with steroid administration and the timing of thymectomy during maturation. An understanding of the pathophysiology of MG in the growing pediatric population is desirable for selecting treatment. Myasthenia caused by impaired signaling at the neuromuscular junction can be broadly divided into acquired MG and congenital myasthenic syndrome (CMS). The pathophysiology, generally known as MG, refers to acquired MG, in which the AChR at the neuromuscular junction is reduced by immunological mechanisms [39]. Most often, this is caused by anti-AChR antibodies, which will be discussed next. It is said that complement is also involved in this process, which is accompanied by the morphological destruction of the neuromuscular junction. On the other hand, CMS is a genetic pathophysiology that is caused by a defect in the production of a protein involved in signal transduction at the neuromuscular junction.

#### 3.1. Formation of Neuromuscular Junction

The neuromuscular junction must be well-formed for this neuromuscular signaling to be rapid and reliable. In denervated rat muscle, which was once used as an antigen in the measurement of AChR antibodies, denervation significantly alters the distribution of AChR in the postsynaptic membranes. There are two types of AChRs: the fetal type and the adult type. The adult type AChR consists of $\alpha$, $\beta$, $\varepsilon$, and $\delta$-subunit, while the fetal type consists of $\alpha$, $\beta$, $\gamma$, and $\delta$-subunit. Both are transmembrane proteins with an ion channel [40]. Surgical denervation and/or presynaptic blockade of neuromuscular transmission amplifies the subunit mRNA of junctional and extra-junctional AChR and increases both types of AChR in muscle cells [40]. Extra-junctional AChRs produced and distributed at the denervated post-synapses are reported to have $\gamma$-subunits instead of $\varepsilon$-subunits, as described above [41].

AChR proteins synthesized in muscle cells and extensively deployed and seeded in the postsynaptic membrane must be assembled at specific locations to form neuromuscular junctions. This requires stimulation from nerves, and a number of proteins are involved. (Figure 2) Nerve terminals secrete agrin, which binds to LRP4 and activates the MuSK molecule. Furthermore, the complex forms a dimer and activates Dok-7 in muscle cells. When the AChE/ColQ protein binds to the MuSK protein molecule, the complex is further activated and stabilized, and AChRs that were widely and thinly distributed in the surrounding area are gathered in the vicinity of MuSK to form neuromuscular junctions [42–44].

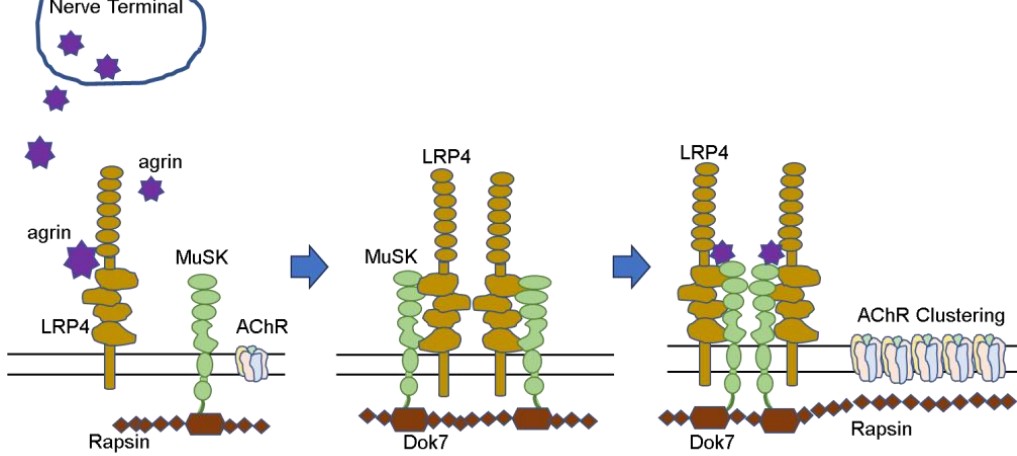

**Figure 2.** AChR clustering (Hayashi. No To Hattatsu 2022. [22]).

In addition to these, several other proteins have been reported to be involved in the formation of this AChR group. When these proteins malfunction, neuromuscular communication is impaired. Mutations that prevent the formation of normal protein molecules and, thus, impair neuromuscular communication are called CMS. The so-called acquired MG is mainly caused by autoantibodies that disrupt signal transduction at the neuromuscular junction, resulting in MG symptoms. The AChR and MuSK antibodies can be clearly identified as the causes of MG symptoms because the method of measurement is well established, the antibodies have been proven in many patient sera, and the same condition can be reproduced in animals by passive transfer. Recently, Lrp4 and agrin antibodies have been added to this group. Autoantibodies against various other proteins involved in neuromuscular junction formation are still under investigation. Sooner or later, it may become possible to identify genetic abnormalities associated with such proteins and to identify autoantibodies against them in patient sera.

### 3.2. Changes at the Neuromuscular Junction

#### 3.2.1. Decreased AChR at the Neuromuscular Junction

Synapses at the neuromuscular junctions are inherently narrow. Degranulated acetylcholine molecules diffuse to the postsynaptic membrane on the opposite muscle side and then bind to AChRs present at the postsynaptic membrane. In 1973, Fambrough et al. took muscle biopsies of eight MG patients and measured the AChR density around the neuromuscular junction [39]. They reported that the AChR density was reduced by 11–30% in MG compared to normal subjects. In the same year, Patrick et al. demonstrated that this disease was an autoimmune disease by immunizing rabbits several times with AChR extracted from the electric organ of Torpedo and reproducing pathophysiology with symptoms of muscle weakness similar to those in humans [2]. In 1974, Almon et al. reported that the binding activity of $\alpha$-bungarotoxin to AChR was decreased in at least 5 of the 15 MG patients when patient serum was added to AChR extracted from rat leg muscles [45]. In 1975, Bender et al. morphologically demonstrated that $\alpha$-bungarotoxin bound to the muscle tissue of MG is eliminated by reacting to patient serum [46]. This indicates that the AChRs present in the postsynaptic membranes react with the patient's serum, causing their density to decrease at the MG patient's own neuromuscular junction. Thus, a series of studies on the pathogenesis of MG was published in the mid-1970s, establishing that MG is an autoimmune disease caused by antibodies in the patient's blood acting to disrupt signaling at the neuromuscular junction.

#### 3.2.2. Structural Destruction of the Neuromuscular Junction

Engel et al. reported that normal neuromuscular junctions have narrow synaptic clefts and well-constructed synaptic folds. In contrast, in MG, the synaptic cleft is enlarged, the synaptic folds disappear, and the debris floats in the enlarged synaptic cleft [47]. This is caused by complement reactions to the autoantibodies described below.

### 3.3. Autoantibodies against the Neuromuscular Junction

In acquired MG, autoantibodies against the neuromuscular junction are formed, which attack the neuromuscular junction, resulting in impaired neuromuscular communication and muscle weakness. AChR is a membrane protein with a molecular weight of approximately 290,000 that is present on the surface of muscle cells, spans the membrane, and is composed of five subunits of four types: two $\alpha$, $\beta$, $\gamma$ (or $\varepsilon$), and $\delta$ subunits. More than half of the autoantibodies present in patient sera are antibodies that target what is called the main immunogenic region, the $\alpha$ subunit of AChR [48].

There are three possible mechanisms by which AChR antibodies cause the reductions in AChR on postsynaptic membranes at the neuromuscular junction in MG, involving binding antibodies, blocking antibodies, and complements [49].

### 3.3.1. Involvement of AChR Antibodies

First, binding antibodies often recognize the α-subunit of AChR, and the Fab portion of the antibody binds to the AChR and bridges the two AChRs, thereby accelerating their uptake into muscle cells and increasing their decay rates [49–51]. The ratio of decayed and lost AChRs to newly produced AChRs determines how much density of AChRs is represented on the muscle cell surface. Generally, AChR antibodies are binding antibodies.

Second, the blocking antibodies recognize and bind to the ACh binding site or its vicinity in the α-subunit of AChR. This prevents ACh from binding to AChR [49,52,53]. The α-subunit is a linear protein consisting of 437 amino acids, each of which has various electric charges and, thus, assumes a three-dimensional structure that penetrates the muscle cell membrane four times. The N-terminal 210 amino acids are completely outside the membrane, and Ach, with a molecular weight of 146, and antibodies, with a molecular weight of approximately 150,000, react with that part of the subunit. AChR is composed of two α-subunits and one each of β, γ (or ε), and δ. The ACh binding sites are located at the contact sites of the $\alpha/\gamma$ (or $\alpha/\varepsilon$) and $\alpha/\delta$ subunits [54].

Third, there is a mechanism by which the complement acts to disrupt the morphology of the neuromuscular junction [47,55]. The morphological changes at the neuromuscular junction caused by the complement are characterized by a wide synaptic gap and a coarse distribution of AChRs, which allows for little information exchange [56]. When the complement repeats the reaction up to C9, it forms a membrane attack complex (MAC) and destroys the membrane [55]. When the presence of the complement is confirmed at the neuromuscular junction [57], and the complement component C3 is removed using snake venom, this synaptic destruction is no longer seen and symptoms improve [58].

### 3.3.2. Neonatal Transient Myasthenia Gravis and Fetal Myasthenia

Some infants born to MG mothers develop neonatal transient MG [59–62]. IgG antibodies are also transferred to the fetus via the placenta except for subclass IgG2, whereas most AChR antibodies are IgG1 and IgG3. Therefore, if the mother's antibody titer is high, the antibodies are naturally transferred to the fetus, and symptoms should appear. However, only 10–12% of newborns develop MG, and most are asymptomatic despite having AChR antibodies in their blood [63]. Idiopathic thrombocytopenic purpura, in which maternal autoantibodies are similarly transferred to the fetus and cause postnatal symptoms, also occurs at a frequency of 20% [64] to 17.8% [65]. However, it is not well understood why neonates do not develop the disease.

This neonatal transient MG declines over time because it does not occur due to antibodies produced by the child on its own but to the transplacental transfer of maternal antibodies. Even if symptoms appear in the first few days of life, they gradually subside with continued treatment and management during the weeks of symptoms.

In rare cases, if the maternal antibodies are specific, transfer to the fetus in high concentrations, and are strong enough to inhibit movement in utero, the fetus may develop a pathophysiology called congenital joint contracture (arthrogryposis congenita) [66,67]. There exist antibodies against the γ-subunit of fetal AChR, which inhibit the response of ACh to its receptor, slow down the ion channel response, and impair signal transduction at the neuromuscular junction. Moreover, a monoclonal antibody (mAb 131) against fetal AChR with such a function has been reported [68].

### 3.3.3. Antibody against Muscle-Specific Tyrosine Kinase (MuSK)

When AChR antibodies are negative, the disease is called seronegative MG. In 2001, Hoch et al. reported that MuSK antibodies are present in 70% of seronegative MG, which accounts for 20% of generalized MG [69]. McConville et al., in the same group, reported that in 66 patients with seronegative MG, 27 (41%) were positive for MuSK antibodies, 11 of whom had prominent bulbar symptoms [70]. Several subsequent reports have shown that about 20% of Caucasian MG patients have seronegative MG, of which 30–40% of the generalized type are positive for MuSK antibodies [71,72]; 38% of adult seronegative MG

cases were reported to be positive for MuSK antibodies by the Mayo Clinic [73]. On the other hand, a survey of adult MG in Asia showed that 21% of generalized type seronegative MG in South Asia [74], 26.4% in China [75], and 26.7% in Korea [76] were positive for the MuSK antibody.

It was reported that MuSk antibodies were positive in 12 (6.7%) of 180 patients who underwent antibody assays with childhood MG with onset before 14 years of age in China, but the details are not clear [25], and we have not been able to find any reports of pediatric MuSK-MG from Korea. There is a report from Taiwan stating that only 1 of 36 AChR antibody-negative patients with juvenile MG was positive for the MuSK antibody [18]. In Japan, Ohta et al. reported that 27% of 85 patients with generalized type seronegative MG had MuSK-MG [77], but the epidemiology of pediatric MuSK-MG is poorly investigated, as only case reports of pediatric cases can be found [78,79]. Thus, subtle differences in the frequency of the disease exist according to region and race.

However, in a report looking at the relationship between MuSK-MG and HLA, Niks et al. reported that MuSK was correlated with HLA-DR14-DQ5 in Dutch people [80], and Kanai et al. reported that HLA-DRB1*14 and DQB1*05 were correlated in Japanese people [81]. Subsequently, Hong et al. reported that HLA-DQB1*05, DRB1*14, and DRB1*16 were correlated in a meta-analysis including the reports of Niks and Kanai, indicating that common HLA types are involved beyond racial differences [82].

The subclass of MuSK antibodies consists mainly of IgG4, which, like AChR antibodies, crosses the placenta to cause symptoms. Neonatal transient MG has been reported from mothers with MuSK-MG [83]. It is rare in MuSK-MG in the ocular muscle type [84,85].

MuSK is responsible for the assembly of AChRs on the postsynaptic membrane in collaboration with several protein molecules to facilitate efficient signal transduction at the neuromuscular junction. (Figure 2) MuSK antibodies have an adverse effect on AChR assembly and reduce functional AChRs, but histopathology does not confirm the loss of AChRs [86]. While the main subclasses of AChR antibodies are IgG1 and IgG3, MuSK antibodies are mainly the IgG4 subclass and IgG monovalent, which crosses the placenta but has no complement binding properties. Konectzny et al. examined the sera of 14 MuSK-MG patients and found that the MuSK antibodies were predominantly IgG4 monovalent with some IgG1-3. MuSK antibodies do not cause the intracellular uptake of MuSKs but instead impair agrin-induced AChR assembly, resulting in MG symptoms [87,88].

The most common sites of symptoms of MuSK-MG are the face, neck, articulatory swallowing, and respiratory muscles, where muscle weakness and atrophy occur. However, it has been suggested that the neuromuscular junction structure of these muscles may be different from that of others or that the low expression of MuSK in the scapulohyoid muscle may result in different responses [89]. MuSK-MG presents with bulbar symptoms, which may become more severe with anticholinesterase agents, whereas the thymus gland is normal, and therefore, thymectomy is not generally performed. Thus, the pathophysiologies of MuSK-MG and AChR antibody-positive MG are different.

MuSK, a protein present at the neuromuscular junction, is essential in order for AChRs to assemble at the neuromuscular junction; when MuSK is deficient or when antibodies block MuSK's natural function, AChRs fail to form clusters, resulting in the inefficient transmission of information from the nerve and the development of MG pathophysiology. Subsequently, several antibodies other than AChR and MuSK antibodies have been shown to cause MG.

### 3.3.4. Double or Triple Seronegative MG

When the AChR antibody is negative, it is called seronegative MG, but when both the AChR and MuSK antibodies are negative, the name "double seronegative MG" is used. When the LRP4 antibody is also negative, "triple seronegative MG" is used. Rodriguez Cruz et al. reported that of 42 MG patients considered to be double seronegative by the immunoprecipitation method, 16 (38.1%) were positive for AChR antibodies according to cell-based assay [90]. In order to test negative for AChR antibodies, it may be necessary to

confirm the result not only by immunoprecipitation but also by cell-based assay. Recently, a study has been reported to measure MuSK antibodies using a cell-based assay [91], but this is still in the research stage and needs to be accumulated in the future. In the same paper reported by Rodriguez Cruz, 26 patients were also considered negative by the cell-based assay method, and LRP4 antibodies were negative in all 21 patients who could be tested [90]. However, Pevzner et al. reported that serum from 12 of 13 double-negative MG patients showed protein deposition at the neuromuscular junction in mice, and in 4 patients, AChR assembly on cultured muscle cells was suppressed by more than 50% [92]. Higuchi et al. reported that 9 of 300 AChR antibody-negative patients had Lrp4 antibodies, and 3 of these 9 patients were also positive for MuSK antibodies [93]. LRP4 antibody-positive MGs are present among double seronegative MGs, but they vary from 2–45% depending on the region [94,95].

Recently, antibodies against agrin have also been investigated, and agrin antibodies have been detected in double and triple seronegative MGs by measuring AChR, MuSK, and LRP4 antibodies [96–99]. LRP4 and agrin, together with MuSK, play a major role in the formation of AChR clusters. The presence of these antibodies prevents the complex formation of MuSK and LRP4, which in turn inhibits AChR assembly and disrupts signaling at the neuromuscular junction.

In general, four conditions are needed to determine whether autoantibodies are responsible for the disease [100,101]. Regarding AChR antibodies, all of these conditions are satisfied: (1) antibodies can be identified in patient sera, (2) the passive immunization of patient sera causes characteristic pathophysiology, (3) the active immunization with antigens causes disease, and (4) the removal of antibodies improves symptoms.

Looking at antibodies against MuSK, LRP4, and agrin for conditions (1) through (3), Shigemoto et al. induced MG symptoms in rabbits by immunizing them with MuSK protein. Pathophysiology showed a reduction in AChR clusters at the neuromuscular junction [102], while Viegas et al. created a MuSK-MG pathophysiology in mice by active immunization with the MuSK protein as well as passive immunization with the MuSK antibody [103].

Similarly, Shen et al. actively immunized mice with the extracellular domain of the LRP protein to induce MG symptoms and passively immunized mice with serum from rabbits immunized with the LRP4 protein to induce the same MG symptoms [95]. Ulsoy et al. [104] and Mori et al. [105] also observed MG symptoms in mice immunized with LRP4 and created autoimmune animals. LRP4 antibodies belong mainly to IgG1 and have complement activity [93].

Yu et al. suppressed MuSK phosphorylation and AChR assembly by passively immunizing mice with immunoglobulin generated from MG patient sera with LRP4/agrin antibodies [106], and Yan et al. immunized mice with agrin to induce the development of MG symptoms [98].

The pathophysiology of acquired MG is thought to be caused by the formation of autoantibodies against protein substances at the neuromuscular junction. The autoantibodies that are currently recognized are against AChR, MuSK, Lrp4, and agrin. All four auto-antibodies appear to meet all of the strict criteria.

There are many other proteins involved in AChR assembly, and several antibodies against these proteins have been identified. It remains to be verified whether these antibodies are really involved in the pathogenesis of the disease.

### 3.4. Congenital Myasthenic Syndrome (CMS)

CMS is characterized by pathological muscle weakness and fatigability caused by an inborn defect of a protein molecule at the neuromuscular junction and is usually diagnosed at the age of 2 years or younger. In addition, CMS is often associated with muscle atrophy and small deformities, so it is important to distinguish CMS from MG as well as from muscular dystrophy and congenital myopathy. Repeated nerve stimulation is essential for definitive diagnosis, and careful and repeated nerve stimulation is desirable in cases of muscle weakness and atrophy without elevated CK [107].

A report from the U.K. showed a high prevalence of 1.5 per million for childhood-onset autoimmune MG in those aged 18 years or younger, compared with 9.2 per million for CMS [108]. The Mayo Clinic reported an incidence rate of 1.2 per million for autoimmune MG and 2.3 per million for CMS in childhood-onset MG under 19 years of age [23]. No epidemiologic reports of CMS have been seen from Asia, only sporadic reports. From Japan, Azuma et al. reported 4 patients with ColQ abnormalities and 5 of their mutations and 5 patients with AChR abnormalities and 6 of their mutations, for a total of 9 patients and 11 of their mutations; however, underdiagnosis is considered highly likely [109]. Thus, to compare the frequency of childhood MG between Europe and East Asia, autoimmune MG is more common in East Asia and less common in Europe and the United States. Conversely, CMS shows a contrasting pattern of onset, being more common in the West and less common in East Asia, with racial differences also present.

CMS is a phenomenon caused by genetic abnormalities in various proteins at the neuromuscular junction that are necessary for neuromuscular signaling to occur, resulting in impaired protein synthesis. Ohno and others have recently reported a review of 35 different genetic abnormalities [110]. The final stage of AChR assembly may involve other unknown proteins, and further studies are needed.

### 3.5. Disease Classification: Ocular and Generalized MG

MG can be broadly classified into ocular MG and generalized MG. In children, ocular MG is common, accounting for 78.2% of cases, and is also known to have low AChR antibody titers and a high percentage of negative titers [111]. In a 2006 epidemiological survey from Japan, 80.6% of cases with onset at less than 5 years of age and 61.5% of cases with onset at 5 to 10 years of age were ocular MG [15]. Sero-negative MG is more common in pediatric patients with ocular MG [111]. Whether or not antibodies are truly absent is a major issue, and Tsujihata et al. confirmed the deposition of anti-AChR antibodies at the neuromuscular junction of the limb muscles by performing limb muscle biopsies of patients with ocular MG [57]. Patients who were thought to have seronegative MG were revealed to have seropositive MG, as antibody titers were detectable against cell-bound AChR due to differences in assay methods [90,112]. In Japan, Oda reported in 1993 that the cell-bound AChR assay using human ocular muscle as an antigen could identify antibodies in ocular MG serum that were negative by means of the radioimmunoprecipitation method [113].

Since the pathophysiology of MG is thought to be caused by autoantibodies, it is thought that even when symptoms are present only in the ocular muscles, antibodies are deposited in other muscles throughout the body, although they are not yet developed due to a threshold for onset. The AChR antibody deposition at the limb muscle's neuromuscular junction, shown above (by Tsujihata), illustrates this idea [57].

In East Asia, ocular MG is more common in children, and most adults also present with ocular muscle symptoms and later develop the generalized type. Why is the ocular MG more common, and why is the ocular muscle more likely to be affected [88,114]? Comparing the neuromuscular junction of the limb muscles with that of the ocular muscles, there are several electrophysiological differences. The ocular muscles require rapid and complex movements of the eyeballs, as well as fixation. Furthermore, the AChR subunits are different, and it is known that $\varepsilon$ is replaced by $\gamma$ in the AChR subunit of the ocular muscle, whereas the limb muscle AChRs originally consisted of $\alpha$, $\beta$, $\varepsilon$, and $\delta$ [115]. The $\gamma$-subunit forms the fetal AChR, and in rodents, many muscles are replaced by the epsilon form within the first week or so after birth [116]. However, this replacement does not occur in the ocular muscles. It has been reported that the neuromuscular junction of ocular muscles has a more complex morphology than that of limb muscles [117], with $\varepsilon$-subunits in simple innervated neuromuscular junctions such as the soleus muscle and $\gamma$-subunits in multiple innervated neuromuscular junctions of the external ocular muscle [116–118]. These differences in the neuromuscular junction accommodate the fine and complex movements of the eye, and the absolute differences in muscle size and thickness create the so-called "safety factor", or margin of safety, which explains why the ocular muscles with less margin

are more prone to symptoms. It has been explained that symptoms are more likely to occur in the ocular muscles with less room to spare [119,120].

### 3.6. Childhood Thymus and Thymic Selection

MG is often associated with thymic abnormalities such as thymoma and thymic hyperplasia. Without distinguishing between thymoma and thymic hyperplasia, Yoshikawa et al. reported that 22.1% of cases were thymoma in Japanese patients [6], and the same was true for 20.1% of Korean patients [11]. Murai et al. reported 32.0% as thymoma and 38.4% as thymic hyperplasia. In MG in children under 9 years of age, thymoma comprised 4.9%, and thymic hyperplasia was 16.8% less common than in adults [15]. Similarly, in China, thymoma and thymic hyperplasia comprised 14.8% and 66.4%, respectively, while thymoma made up 2.9% and thymic hyperplasia 86.5% in pediatric MG cases aged 14 years or younger [16]. Popperud et al. reported that 50 of 63 pediatric MG patients under the age of 18 had undergone thymectomy. Of the 21 prepubertal patients, 13 had thymectomy and 7 (54%) had thymic hyperplasia, and of the 42 postpubertal patients, 37 had thymectomy and 23 (62%) had thymic hyperplasia. No thymoma was reported either before or after puberty [21], and Heckman et al. also reported that thymoma is rare in children [121]. Thus, the critical difference is whether the thymus is a tumor or hyperplasia in both children and adults. If it is a thymoma, removal is the treatment of choice.

The thymus gland is an organ that all humans are born with, and it plays a major role in the development of immunity that is able to distinguish between self and non-self [122,123]. If it were not present, we would have primary immunodeficiency.

The thymus gland usually atrophies after completing its work of establishing immunity at a young age, and most disappear by about age 40 [122,124,125]. Therefore, if thymectomy is indicated in children, it should be performed after puberty. Thymoma, which occurs more frequently in adults, has a different implication. Since the thymus gland, which should be atrophied, is instead enlarged and harms the person as a tumor, its removal is considered to be the appropriate treatment.

### 3.7. Involvement of Cellular Immunity

The involvement of HLA was mentioned above as an influence of one's immunogenetic background. In particular, HLA-DR is MHC-class II, which expresses antigen peptides bound to its pocket on the surfaces of antigen-presenting cells and reacts with the TCR of T cells. As an experimental model, Berman and Patrick examined the murine system using various types of mice [126]. The results showed that C57BL/6(B6), which is Th1-dominant, is used as an experimental model for MG instead of Balb/c, which is Th2-dominant. The mouse MHC, H-2 complex, is haplotype-b in B6 and haplotype-d in Balb/c. Christadoss et al. have shown that the differences in MG-susceptibility between species of mice are due to differences in T-cell activity in response to the same antigenic stimuli and that the MG-susceptibility of the H-2 gene is linked and that it is controlled by the I-A subregion of the MHC linked to the H-2 gene [127].

McIntyre and Seidman have created a B6.C-H-2bm12 (bm12) mouse with three mutations in the I-Aβ of the B6 mouse [128]. This bm12 mouse, with only three mutations in I-A, had a greatly reduced incidence of MG development [129]. This is reported to be due to a significant effect on the epitope repertoire of murine CD4+ T-cells sensitized to AChR [130,131]. Wu et al. created systemic MG by immunizing HLA-DR3 transgenic mice with human AChR epsilon-subunit [132].

We reported a case of a patient who developed chronic myeloid leukemia (CML) and received an HLA-matched bone marrow transplant from her sister, who was doing well but developed graft-versus-host disease (GVHD) and then MG due to neglected medications during treatment [133]. CML treatment with intense chemotherapy and radiotherapy may result in thymic damage, which may lead to the release of abnormal lymphocytes into the peripheral circulation and the development of autoimmunity. GVHD is said to be caused by an immune imbalance due to decreased Tregs [134].

MG is also more likely to occur when autoimmune diseases are present in the family. The frequency of MG increases markedly in twins [135]. These events may indicate that MHC antigens expressed on antigen-presenting cells are transmitted within the family, making the family susceptible not only to MG but also to other autoimmune diseases.

MG is a T-cell-dependent, antibody-producing autoimmune disease. As such, treatment has included steroids, immunosuppressive agents, and, more recently, various biological agents, which are more easily implemented in treatment and more effective. However, the use of such drugs in children is limited by the difficulty of conducting clinical trials to confirm their safety [136,137]. Immunosuppressive agents such as azathioprine and cyclophosphamide have been used for severe cases in Japan, but they are not actively used because they are not covered by insurance. Currently, tacrolimus and ciclosporin A are the two calcineurin inhibitors that are covered by insurance, and eculizumab, a biologic, has recently become available for use in pediatric patients. Eculizumab is a C5 inhibitor of complement that prevents MAC (membrane attack complex) formation and the destruction of neuromuscular junctions. The calcineurin inhibitor, on the other hand, is quite effective, although it does not act directly on antibody production but rather suppresses T-cell activity upstream of it.

### 3.8. Lymphorrhage

MG is a T-cell-dependent disease in which antibody production occurs and neuromuscular signaling is disturbed. Autoantibodies against various proteins involved in AChR assembly, mainly AChR antibodies, are thought to be involved in this pathophysiology, including anti-MuSK, anti-LRP4, and anti-agrin antibodies. However, seronegative MG actually exists and accounts for 10% of generalized MG [138]. Oda fixed ocular muscle AChR to wells and measured AChR antibody titers in the serum of patients with ocular MG but reported that some cases were still negative [113]. A comparison of soluble IL2 receptor (sIL2R), a marker of T-cell activation, with AChR antibody titers showed a significant negative correlation, with lower antibody titers resulting in higher sIL2R [139]. This result suggests that T-cells may be activated in MG cases with low AChR antibody titers. In ocular MG, antibody titers are low, and T-cells may be activated. In cases where a patient is diagnosed with seronegative MG and no antibody is found, no matter the activity of the known antibodies, are there still antibodies that have not been found? Or are they just antibodies that are below the sensitivity of measurement? Lymphorrhage used to be a major issue in the days when many MG patients died and were autopsied [140], but since then, there has been controversy regarding the existence and pathological significance of lymphorrhage. In 1963, Fenichel et al. performed muscle biopsies on 37 MG patients and divided the tissues into three groups: 15 cases in the normal group, 11 cases in the small muscle group showing denervated muscle, and 12 cases in the lymphocytic infiltration group. Most of the biopsied muscles were quadriceps muscles, and the time from onset to biopsy was 1 to 8 years in most cases. Lymphocytic infiltration was seen at a high rate in muscle biopsies as well as autopsies and was more common in cases with a short period of time after onset and thymic abnormalities [141,142]. Pascuzzi and Campa biopsied the tricep muscles and found lymphorrhage in the muscle endplate, pointing to the possibility that cellular immunity is involved in the pathogenesis of MG [143]. Furthermore, Maselli et al. biopsied the anconeus muscles of eight MG patients and found cellular infiltration of the neuromuscular junction in seven patients [144]. On the other hand, Nakano et al. found inflammatory cell infiltration around the endplate in 12 of 30 patients, but the degree was mild, less than 10% of the endplate. They claimed that lymphorrhage was a nonspecific phenomenon [145]. Most of the muscle biopsy sites were external intercostal muscles.

In studies producing MG pathology in experimental animals, infiltration of lymphocytes around the neuromuscular junction has been seen in the acute phase, which occurs after antigen injection in rats. In the chronic phase, when AChR antibody titers rise, this lymphocytic infiltration disappears and morphological destruction of the neuromuscular junction is observed [146]. The same phenomenon is seen in the immunization of

extracted and purified AChR [146], as well as in the passive transfer of AChR antibodies in serum [147–149].

In the mid-20th century, steroids began to be used, and a dramatic change in treatment occurred. Until then, there was no treatment available to deal with the rapidly progressing pathology of MG, so muscle tissue was viewed during autopsy at the end of the natural course of the disease. The literature of the time recorded the pathological findings in detail. The advent of anti-cholinesterase agents, ACTH, and thymectomy extended the clinical course of the disease. The use of steroids has saved many lives, deaths from MG have almost disappeared, and muscle tissue specimens are often obtained from localized muscle tissue such as the pectoralis major and intercostal muscles at the time of thymectomy. As studies using experimental models have shown, the pathophysiology is variable. The scene of the pathophysiology presented by the patient may differ depending on the time since the onset of the disease, and the judgment of the pathophysiology may differ depending on the object being viewed. The classic literature has repeatedly shown the importance of lymphorrhage. We believe that in some cases of seronegative ocular muscle type, lymphorrhage may occur in the ocular muscles, leading to the onset of the disease.

## 4. Conclusions

Recent progress in epidemiological studies of childhood myasthenia and treatment methods has allowed for the collection of reports from various regions of the world. The results of these studies have revealed subtle differences in the frequency of autoimmune MG and CMS between racial and regional groups, as well as in the pathophysiology of myasthenia and treatment methods, based on the history of medical care in each country and region.

Research on the pathophysiology of myasthenia has made great strides in the past half century, and there are now a variety of treatment options available, including the administration of immunosuppressive and biologic preparations. This review did not address treatment but outlined the pathophysiology of myasthenia in general. A thorough understanding of this condition is important for its treatment. Children differ significantly from adults in that they are growing and developing. The frequency of childhood MG is clearly lower than that of adult-onset MG, and the basic diagnosis and treatment are often chosen to mimic those of adult MG, assuming that the pathophysiology of the disease mirrors that in adults. However, it is clear, as mentioned above, that there is a pathophysiology unique to pediatric MG, and treatment methods should be selected accordingly. This paper has discussed some of the ways in which pediatric MG differs from adult MG. However, the reason for these differences still needs to be investigated. Immunogenetic background and the development of the nervous and immune systems may play a role. I believe that we must advance research in this unresolved field.

**Funding:** This study received no external funding.

**Institutional Review Board Statement:** Not applicable for this review, which did not involve human or animal studies.

**Informed Consent Statement:** Informed consent was obtained from the patient.

**Data Availability Statement:** Not applicable.

**Conflicts of Interest:** The author declares no conflict of interest.

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
