# Peer review of "Pathophysiology of Childhood-Onset Myasthenia: Abnormalities of Neuromuscular Junction and Autoimmunity and Its Background"

_pathophysiology, doi:10.3390/pathophysiology30040043_

Round 1

Reviewer 1 Report

Comments and Suggestions for Authors

I think this is a competent effort and a useful review for the physicians looking at the patients with child-onset MG.

I have a few comments that should improve manuscript.

(1)  Author should meniton the data based on child-onset MG in the title of Table 1.   

(2)  The image of Figure 3 is familiar with most of MG specialists. If author insert Figure 3, I would recommend to get reprint consent the authors and JCI (Reference 46 Conti-Fine BM, et al).

(3)  The last section entitled “3.7.1 Lymphorrhage” gives a sudden impression. Author needs reconstructing along with the revious section entitled “ 3.7. 

Comments on the Quality of English Language

Minor editing English language required

Author Response

Reply to the Reviewer 1,

Thank you for your valuable comments. This is a long review article and must have taken a long time and effort to finish reading. I thank the reviewer for commenting on it. Based on your comments, I would like to change my mauscript. The rewritten portions are noted in red.

(1) You commented to mention the data based on child-onset MG in the title of Table 1. According to your comment, I changed the title of the Table 1. The content of Table 1 is described (Line 91-108).

   Comparison of childhood-onset MG: ocular MG is often in Asia

(2) You commented how to use Figure 3. As you commented, the content of Figure 3 may be familiar to many pople associated with MG. I would like to omit this Figure 3, according to your suggestion.

(3) You commented about Lymphorrhage. Basically, MG is an autoimmune disease due to autoantibodies such as AChR Ab and MuSK Ab. But, I believe that there remains a possibility of cell-mediated immunity in some cases for which antibodies cannot be identified by any method. I would like people to reaffirm this way of thinking and make it the subject of their research. Since Lymphorrhage is also about cellular immunity, I summarized it as a chapter in "Involvement of cellular immunity". But, as pointed out in the review, I would like to summarize it as a separate chapter as 3.8.

Reviewer 2 Report

Comments and Suggestions for Authors

Dear Author, 

i read with great interest your manuscript entitled "Pathophysiology of childhood-onset myasthenia: Abnormalities of neuromuscular junction and autoimmunity, and its background".

Although this paper is not scientifically sounds, it is a well-written overview about childhood onset MG and It may be useful for clinicians and researchers providing a easy and rapid overview of childhood MG thus could be published. 

Kind regards

Author Response

Reply to the Reviewer 2,

Thank you very much for your positive evaluation of this paper. You must need long time and much effort to read out this long manuscript. Thank you so much again.

Reviewer 3 Report

Comments and Suggestions for Authors

The presented review aims to focus on the peculiarities of myasthenia's pathogenesis in children. As paediatric/juvenile MG is more common and distinctly different in presentation the East Asian race, the attention expectedly turns to studies on this population. The author is an expert in the field with publications on the subject over the last decades and I expected to find some interesting synthesis of his expertise and literature data. Indeed, in this lengthy paper we find abundant information, systematised into chapters on epidemiology of MG, autoimmunity etc. However, among this abundance the main subject of pathogenesis of juvenile MG and how and why it differs from the adult seems at times neglected.

I would propose improvements to make the paper better readable; I have some conceptual questions as well.

1. Concerning the whole piece, I would ask the respected Author to edit it into a much more compact narrative. Throughout the text one finds tautologies, redundant phrases, verbose explanations etc. A random example: "The gap between nerve endings and muscle, called the synapse at the neuromuscular 217 junction, is inherently narrow". I presume the audience of the Journal "Pathophysiology" is aware what is a synapse and the explanation is not necessary. At times, whole paragraphs are dedicated to detailed descriptions of some phenomena or , that may be summarised in a few sentences. 

2. Some of the terminology used is not standard. I would point at "ocular muscle type myasthenia" that is called in all sources "ocular myasthenia", the use of "pathophysiology" as a synonym for "dysfunction, disturbance, pathological mechanism" (e.g .  "...abnormalities in miniature endplate potential (MEPP), which they identified as a pathophysiology of the neuromuscular junction...").  The clusters of AChR are also called groups, aggregates, assemblies - which is not wrong by itself, but should not replace the standard term for these molecular structures. 

To summarise, I believe some style edition to make the paper more concise  would improve much its readability.

I have some particular questions. Starting from 1) Introduction - in this brief historical excursion I wonder why J Simpson is omitted as the proponent of autoimmune theory of myasthenia graves. Elmquist seems to have published the observation on small MEPP already in 1965  (Elmqvist D. Neuromuscular transmission with special reference to myasthenia graves. Acta Physiol Scand Suppl. 1965:SUPPL 249:1-34. PMID: 14298289).

2. Epidemiology. Immunogenetic. Gender difference. - In this chapter, may I propose mentioning the division of juvenile MG into pre-, peri- and post pubertal, with the corresponding differences in gender involvement, from about 1.3 Female/Male in prepubertal to 4 or more/1 in post pubertal? Maybe a comment on the very definition of "childhood MG" that actually includes prepubertal children along with post pubertal adolescents would be welcome.

3. Pathophysiology... - in this lengthy chapter, the detailed explanations seem quite sufficient. An useful addition to the discussion would be to include the methods for detection of the antibodies in the discussion as an explanation for part of the controversies around seronegativity and the true percent of MG due to different antibodies; I mean the newer sensitive methods like detection of Ab against clustered receptors and the cell based assays. Also, commenting on MuSK MG, one could expand on the mechanism of pathogenicity.    In 3.3.4. mentioning the "rules of Kaminski" - the rules were actually formulated by Drachman (Drachma DB. How to recognize an antibody-mediated autoimmune disease: criteria. Res Publ Assoc Res Nerv Ment Dis. 1990;68:183-6. PMID: 2183310).  

Finally in the chapters on possible direct T-cell influence on the neuromuscular synapse it may be mentioned, that support on this possibility is mostly indirect, and while the hypothesis of direct damage of the synapse by Т-cells still exists and may explain some rare seronegative cases, the prevailing opinion is that Myasthenia is due to antibody-mediated disturbance mostly of the postsynaptic membrane.

I hope the author will accept my well-intended notes especially regarding the style.

Comments on the Quality of English Language

I would suggest a professional language edition, as a number of awkward phrases and sentences are encountered in the texts that at time interfere with the meaning. Eg "MuSK-MG ...is 344 easily severe,(?), ..."when MuSK is deficient in a pathophysiology (?)...' 

Author Response

Reply to the Reviewer 3, Thank you for your valuable and detailed comments. As the reviewer also mentioned in their comments, it is a long review article and must have taken a long time and effort to finish reading. I thank the reviewer for commenting on it in detail. Based on your comments, I would like to change my manuscript. The rewritten portions are noted in red. (1) First, there was a comment that it is ignored as to why and how pediatric MG differs from adult MG. Due to the medical situation in Japan, I have been treating both pediatric and adult MG for many years while continuing to practice pediatrics. I realize that children and adults are different in many points. I described various differences between children and adults in the paper. These include the frequency of different ages of onset and racial differences, the ratio of ocular MG to generalized MG, the height of autoantibodies and seronegativity, and the thymus gland. However, it is unknown why these differences occur. We are considering the possibility that immunogenetic background and neurological and immune system development may be contributing factors, but this is an unknown issue. The main reason for compiling this review paper was to draw attention to this unknown part. I mentioned this point at the end of the paper, as follows; This paper has discussed some of the ways in which pediatric MG differs from adult MG. However, the reason for the difference still needs to be investigated. Immunogenetic background and development of the nervous and immune systems may play a role. I believe that we must advance research in this unresolved field. (2) The second major problem was that the sentences were long and difficult to read. I understand that the field of “Pathophysiology” is broad and there are people with various specialties, so I thought it would be better to use an easy-to-understand expression. As you pointed out, I have corrected it to be as compact as possible. And then, I submitted it for English correction. (2-1) â‘  Line 217→226 “The gap between nerve endings and muscle, called the synapse at the neuromuscular junction, is inherently narrow.” has been rewritten. Synapses at the neuromuscular junctions are inherently narrow, â‘¡ Line 183-5 One sentence has been omitted. â‘¢ Lines 201-4→212 One sentence has been omitted and rewritten to be shorter. When these proteins malfunction, neuromuscular communication is impaired. â‘£ Line 257 “old” has been omitted. ⑤ Line 483-97→493 This portion has been omitted, and rewritten to be shorter. The thymus gland is an organ that all humans are born with, and it plays a major role in the development of immunity that is able to distinguish between self and non-self. If it were not present, we would have primary immunodeficiency. (2-2) I was instructed to use standard Terminology. I have rewritten it as simple and concise as possible. then submitted it for English correction. â‘¥ All expressions such as ocular muscle type are ocular MG or generalized MG, with a few exceptions. ⑦ Regarding pathophysiology, I believe that the MEPP part of Line 13 that you pointed out is a natural expression. â‘§ Regarding the words “clusters”, “aggregation”, and “assembly” that you pointed out, I use “cluster” and “assembly” according to their meaning. Line 36 dysfunction of AChR assembly Line 339, 363 and 391 aggregation→assembly, Line 350 groups→clusters Line 381 aggregation→clusters Line 398 clustering→assembly (3) Thank you for pointing out that I should comment on J Simpson. As you pointed out, I added Simpson's work (Line 31). and as expected from clinical findings by Simpson in 1960, it became clear that MG is an autoimmune disease of the neuromuscular junction. (4) You also pointed out Elmqvist. According to your suggestion, I looked for the Acta Physiol Scand Suppl 1965 paper, but I could not obtain it for this short term. But actually I listed the Elmqvist’s paper from 1964, in the first manuscript. The reference is listed as 1964, but I made a careless mistake and wrote it as 1971 in the text. I have corrected it with apologies. (5) In the Gender Differences Chapter, we were advised to mention the prepubertal and postpubertal gender differences. It is stated in the chapter “Gender difference” as such; Several studies have looked at gender differences in the onset of MG. Murai et al. reported that the male to female ratio was 1.6 for cases of onset between 0 and 4 years of age, 1.5 for cases of onset between 5 and 9 years, 2.3 for cases of onset between 10 and 49 years of age, and 1.3 for cases of onset between 50 and 64 years of age. According to a report by Huang et al. from China, this ratio is almost 1 for children with onset of disease under 14 years of age, but there are more women with onset of disease between 15 and 59 years of age, and on the contrary, men with onset of disease after 60 years of age. Furthermore, Finnis et al. reported that in childhood-onset MG, there is no difference between men and women in prepubertal, but this ratio is 4.5 in pubertal and 4.5 in postpubertal. Although there are subtle differences depending on the report, there is almost no difference between men and women in the prepubertal period, but after the pubertal period, there are more women, and as they get older, there are more men. In addition, the definition of childhood in this paper is defined as 18 years of age or younger, and this is stated (Line 94). In this paper, I would like to define childhood as 18 years of age or younger, including post-pubertal age. (5) You commented that it would be better to further mention cell-based assay methods in Pathophysiology. I have already mentioned Rodriguez Cruz's paper in the Seronegative MG chapter (Line 357-359→Line 362-368), but I added a comment after that. Rodriguez Cruz et al. reported that of 42 MG patients considered double seronegative by immunoprecipitation method, 16 (38.1% antibodies) were positive for AChR by cell-based assay. In order to test negative for AChR antibodies, it may be necessary to confirm the result not only by immunoprecipitation but also by cell-based assay. Recently, a study has been reported to measure MuSK antibodies using a cell-based assay, but this is still in the research stage and needs to be accumulated in the future. However, In Japan, there are very few patients with MuSK-MG in the pediatric field, therefore, cell-based assay of MuSK antibodies is still a distant area. I have also mentioned this matter (cell-based assay for AChR Ab) in the chapter of Disease classification (Line 436-440→Line 446-451). Patients who were thought to have seronegative MG were revealed to have seropositive MG, as antibody titers were detectable against cell-bound AChR due to differences in assay methods. In Japan, Oda reported in 1993 that the cell-bound AChR assay using human ocular muscle as an antigen could identify antibodies in ocular MG serum that was negative by means of the radioimmunoprecipitation method. (7) Regarding the Rule of Kaminski, I received a comment that it should be Drachman's rule. Due to my lack of knowledge, I have corrected this point (Line 374→384). In general, four conditions are needed to determine whether autoantibodies are responsible for the disease. Regarding AChR antibodies, all of these conditions are satisfied: (8) I received your comments regarding Lymphorrage. As you pointed out, the main pathology of the disease MG is T cell dependent AchR Ab productive autoimmune disease. However, I believe that there remains a possibility of cell-mediated immunity in some cases for which antibodies cannot be identified by any method. I would like people to reaffirm this way of thinking and make it the subject of their research.

Round 2

Reviewer 3 Report

Comments and Suggestions for Authors

Dear respected Author, thank you for your kind understanding and taking into consideration my humble remarks. I hope they have been of use and I will be happy to see your valuable paper in press now.

Author Response

Reply to the Reviewer 3,

Thank you for your further response and positive comments.

According to the Academic Reviewer's advice, I changed the figure 2 in color. If this paper is accepted and open for the clinicians and scientists, I would be happy.   
